# Three-dimensional aromaticity in an antiaromatic cyclophane

Ryo Nozawa[1], Jinseok Kim[2], Juwon Oh[2], Anna Lamping[3], Yemei Wang[4], Soji Shimizu[4], Ichiro Hisaki[5], Tim Kowalczyk[3], Heike Fliegl[6], Dongho Kim[2] & Hiroshi Shinokubo [1]

Understanding of interactions among molecules is essential to elucidate the binding of pharmaceuticals on receptors, the mechanism of protein folding and self-assembling of organic molecules. While interactions between two aromatic molecules have been examined extensively, little is known about the interactions between two antiaromatic molecules. Theoretical investigations have predicted that antiaromatic molecules should be stabilized when they stack with each other by attractive intermolecular interactions. Here, we report the synthesis of a cyclophane, in which two antiaromatic porphyrin moieties adopt a stacked face-to-face geometry with a distance shorter than the sum of the van der Waals radii of the atoms involved. The aromaticity in this cyclophane has been examined experimentally and theoretically. This cyclophane exhibits three-dimensional spatial current channels between the two subunits, which corroborates the existence of attractive interactions between two antiaromatic π-systems.

[1] Department of Molecular and Macromolecular Chemistry, Graduate School of Engineering, Nagoya University, Nagoya 464-8603, Japan. [2] Department of Chemistry, Yonsei University, Seodaemoon-gu, Seoul 03722, Republic of Korea. [3] Department of Chemistry, Advanced Materials Science & Engineering Center and Institute for Energy Studies, Western Washington University, Bellingham, WA 98229, USA. [4] Department of Chemistry and Biochemistry, Graduate School of Engineering, Kyushu University, Fukuoka 819-0395, Japan. [5] Research Institute for Electronic Science, Hokkaido University, Sapporo 001-0020, Japan. [6] Karlsruhe Institute of Technology, Institute of Nanotechnology, 76344 Eggenstein-Leopoldshafen, Germany. Correspondence and requests for materials should be addressed to T.K. (email: Tim.Kowalczyk@wwu.edu) or to H.F. (email: heike.fliegl@kit.edu) or to D.K. (email: dongho@yonsei.ac.kr) or to H.S. (email: hshino@chembio.nagoya-u.ac.jp)

Cyclophanes are macrocyclic molecules, in which two aromatic subunits are connected in a face-to-face orientation through spacers (Fig. 1a). Chemists have extensively investigated the synthesis, structures, and properties of cyclophanes in order to elucidate the through-space interactions between the aromatic subunits[1,2]. Due to the repulsion between π-electrons, the aromatic rings deviate from their intrinsic planarity, which leads to a destabilization of the cyclophane molecules, especially in cyclophanes with short spacers.

One long-standing issue that remains to be addressed regards the interactions between two antiaromatic π-systems with $4n$ ($n = 1, 2, 3…$) π-electrons. In sharp contrast to cyclophanes composed of aromatic subunits (AR-cyclophanes), theoretical calculations have predicted that the corresponding cyclophanes containing two face-to-face antiaromatic π-systems (AN-cyclophanes) should be expected to be stabilized through mutual orbital interactions between the stacked antiaromatic subunits (Fig. 1b)[3]. Other theoretical investigations have postulated three-dimensional aromaticity in AN-cyclophanes, which would manifest in diatropic ring current effects[4–6]. However, AN-cyclophanes with three-dimensional aromaticity have not been obtained experimentally to date. Synthetic attempts to generate AN-cyclophanes have failed so far, due to the instability and high reactivity of the antiaromatic subunits, as well as the difficulties associated with the synthesis of cyclophanes with very short interplanar distances (Fig. 1c)[7–12]. Thus, evidence on whether attractive interactions exist between the two antiaromatic π-systems in such AN-cyclophanes remains elusive.

Even though antiaromaticity has been known for a long time[13], it has recently attracted particular interest due to its potential attractive features for applications in materials science, e.g. ambipolar carrier mobility[14], the ability to store energy[15], and single-molecule conductivity[16]. Norcorrole 1 is an antiaromatic porphyrin analog[17] that contains 16 π-electrons (Fig. 2a), and we have already reported the synthesis of a stable norcorrole Ni(II) complex[18]. The observed reactivity, optical properties, and electrochemical properties of this complex are markedly different from those of conventional aromatic porphyrins[19,20].

Recently, we have also synthesized a tethered norcorrole dimer 2 (Fig. 2b), which predominantly adopts a stacked structure in the solid state[21], and we have discovered that the antiaromatic character of 2 significantly decreases in the stacked state.

However, the flexible linker in 2 inevitably results in a dynamic interchange between the non-stacked and stacked conformations. The contribution of the non-stacked state disrupts the effective orbital interactions between the two π-systems, and thus dimer 2 remains an antiaromatic molecule.

To achieve perfect face-to-face alignment of two antiaromatic systems, we have designed and prepared rigid AN-cyclophane 5 (Fig. 2c). Such a closely stacked antiaromatic system should be expected to generate distinct three-dimensional aromaticity due to the effective overlap of the molecular orbitals of the two π-systems. Herein, we discuss the electronic and photophysical properties of 5 and evaluate its three-dimensional aromaticity by spectroscopic methods and theoretical calculations.

## Results

**Synthesis of AN-cyclophane.** The design of 5 is based on bithiophene linkers that connect two norcorrole macrocycles. We synthesized 5 using a twofold intramolecular and intermolecular homo-coupling of dipyrrin Ni(II) complex 3 via intermediary norcorrole 4 (Fig. 3a). Treatment of 3 with 5.0 equiv of Ni(cod)$_2$ and 2,2'-bipyridyl in THF at 90 °C afforded 5 in 7% yield, together with a small amount of monomer 6.

The molecular structures of 5 and 6 in the crystalline state were unambiguously determined by X-ray diffraction analysis (Fig. 3b, c as well as Supplementary Fig. 5). The two norcorrole moieties in 5 adopt a closely stacked arrangement. The interplanar distance between the two norcorrole planes (3.09 Å) is considerably shorter than the sum of the van der Waals radii of two $sp^2$-hybridized carbon atoms (3.4 Å), while the distance between the two nickel atoms is only 2.975 Å. A particularly noteworthy feature is that both norcorrole subunits maintain their planar conformation with mean plane deviations of 0.067 and 0.105 Å, respectively. This configuration stands in sharp contrast to that of AR-cyclophanes, in which aromatic subunits are often distorted due to π-electron repulsion[1]. Moreover, the two norcorrole units are rotated relative to each other by merely 22°, which is significantly less rotation than in tethered norcorrole dimer 2 (66°). This orientation should thus enable effective interactions between the molecular orbitals of the two norcorrole subunits. Interestingly, the bond length alternation (BLA) in cyclophane 5 is significantly smaller than that in monomer 6, indicating the

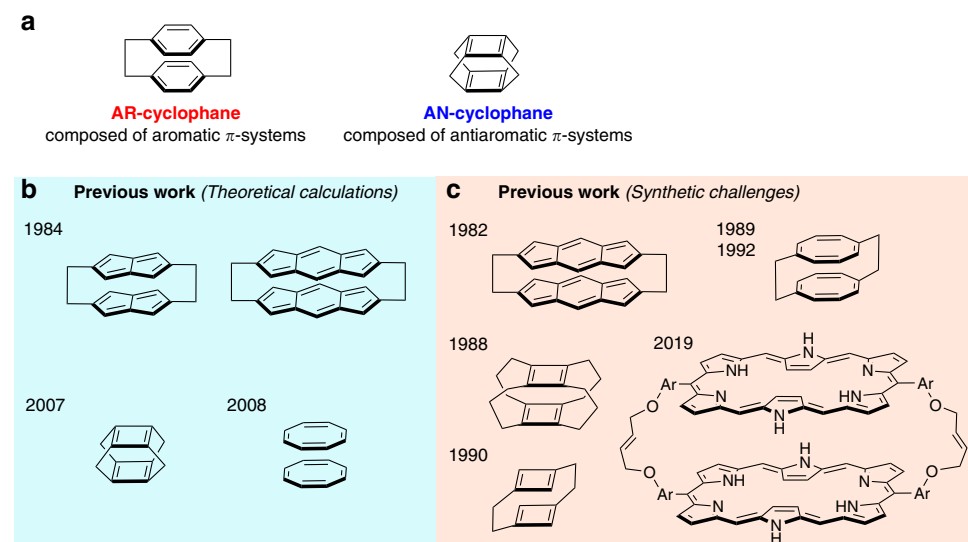

**Fig. 1** Cyclophanes. **a** Representative examples of aromatic cyclophanes (AR-cyclophanes) and antiaromatic cyclophanes (AN-cyclophanes). **b** Theoretical studies on AN-cyclophanes. **c** Attempted unsuccessful syntheses of AN-cyclophanes

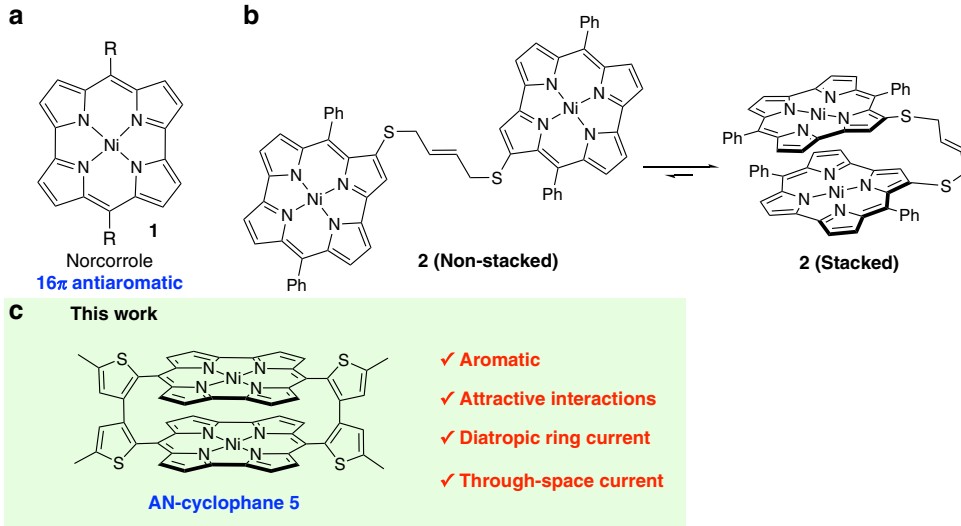

**Fig. 2** Norcorroles and a norcorrole cyclophane. **a** Chemical structures of norcorrole Ni(II) complex **1**. **b** Tethered norcorrole Ni(II) dimer **2** and its dynamic equilibrium. **c** AN-cyclophane **5** which contains two stacked norcorrole Ni(II) units

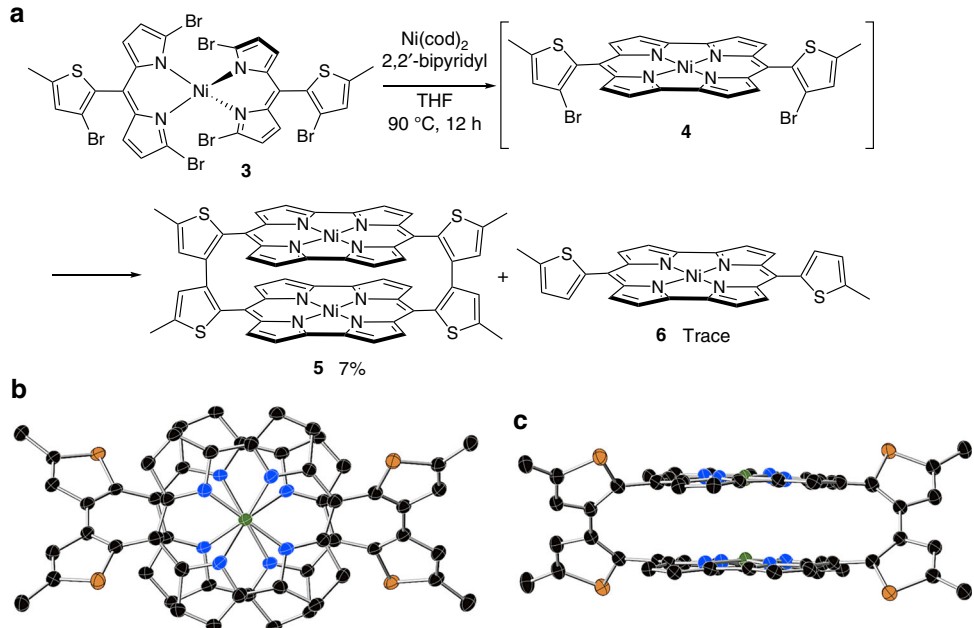

**Fig. 3** Synthesis and structure of **5**. **a** One-pot synthesis of **5** from **3**. **b**, **c** Top and side views of the X-ray diffraction structure of **5** (hydrogen atoms are omitted for clarity; thermal ellipsoids are shown at 50% probability)

more effective delocalization π-electrons in **5**. The BLA can be evaluated using the harmonic oscillator model of aromaticity (HOMA) values[22]. While a HOMA value of **6** is 0.38, those of two norcorrole rings in **5** are 0.57 and 0.54. We also investigated the conformations of **5** using density functional theory (DFT) calculations (Supplementary Fig. 10). While **5** in the solid state adopted nearly $D_2$ conformation, the relative energies of other possible conformers were much higher.

**Ring current analysis**. The ring current in **5** was experimentally evaluated on the basis of its $^1$H NMR spectrum, which exhibits two doublets for the pyrrole protons at $\delta = 5.7$–$6.1$ ppm (Supplementary Fig. 3). These signals are substantially downfield-shifted compared to the corresponding signals of monomer **6** ($\delta = 2.3$–$3.1$ ppm; Supplementary Fig. 4). This phenomenon

clearly indicates the reversal of the inherent paratropic (antiaromatic) ring current of the norcorrole subunits to a diatropic (aromatic) ring current in the closely stacked AN-cyclophane **5**.

To examine the ring current effects in **5** and **6**, we estimated the nucleus-independent chemical shift (NICS) values[23] from their X-ray structures using DFT calculations. The two-dimensional NICS plot of monomer **6** clearly depicts a largely positive region in blue, which supports its strong antiaromaticity (Fig. 4a as well as Supplementary Figs. 14a and 16). On the other hand, the central negative region in yellow and green in the NICS plot of AN-cyclophane 5 indicates the presence of a diatropic ring current, confirming its aromatic nature (Fig. 4b as well as Supplementary Figs. 14b and 16). The three-dimensional plots of the magnetic shielding effect also support that **5** and **6** present opposing ring currents (Supplementary Fig. 15).

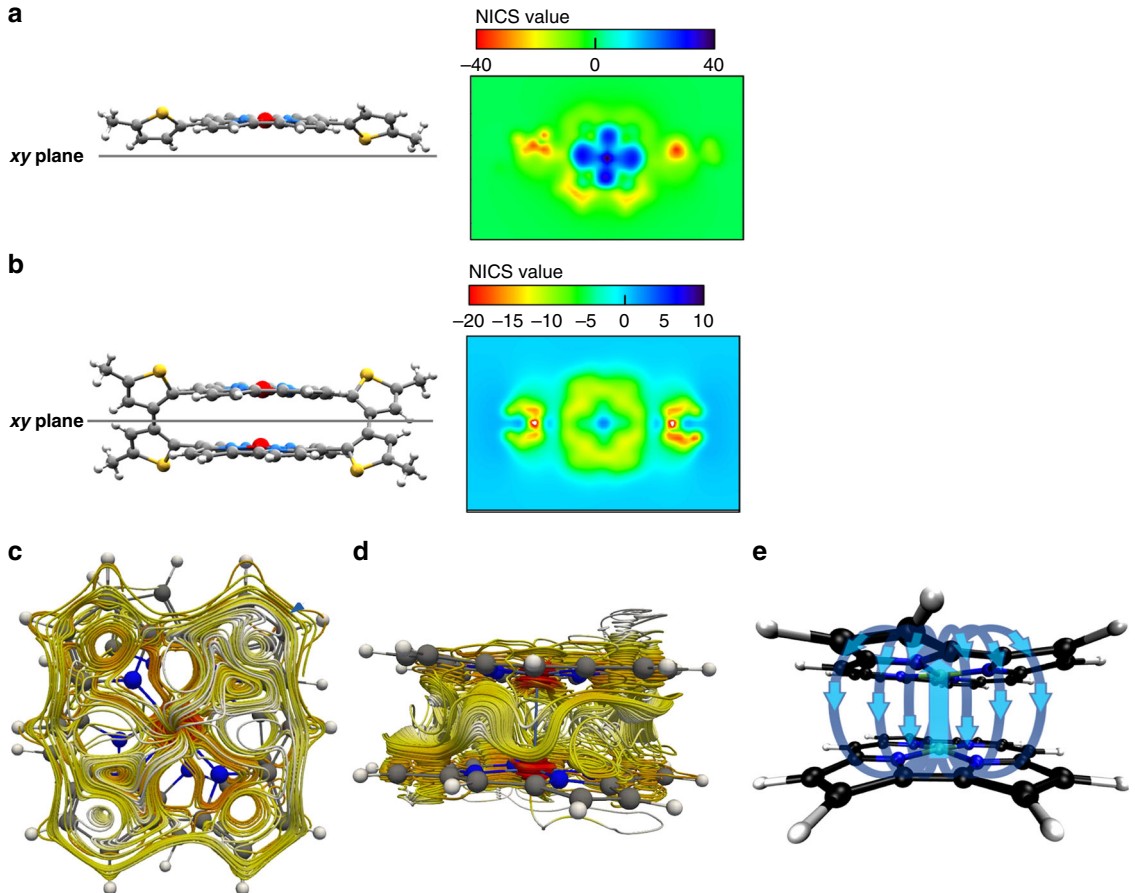

**Fig. 4** Computational results supporting the aromatic nature of AN-cyclophane **5**. **a**, **b** Two-dimensional NICS plots of **6** and **5** in the *xy* plane. **c**, **d** Top view and side view of the calculated magnetically induced current in the model system **5′** obtained using the GIMIC method. **e** Schematic visualization of the through-space current channels in **5′**

To further confirm the aromatic nature of **5**, we performed a current density analysis by the gauge-including magnetically induced current (GIMIC) method[24,25]. The GIMIC method is a reliable means to determine the degree of aromaticity in a quantitative manner when clearly separated current streams are present. Generally, ring currents of aromatic compounds flow on the molecular plane. The current flow in conventional π-systems is restricted to a two-dimensional space. For this analysis, we used a model system **5′** without the bithiophene spacers as well as the real molecule **5**. Streamline visualization of the current density showed that the current pattern for **5′** is rather complex (Fig. 4c). However, a close visual inspection revealed that a clear current channel flows along the Ni–Ni axis and separates afterwards into at least four different current channels that spread over the entire norcorrole ring. The current then travels back through space to the other norcorrole moiety (Fig. 4e). In addition, a weaker S-shaped current channel was identified, which flows between the two norcorrole units around the whole molecule (Fig. 4d). Each norcorrole unit sustains a weak diatropic ring current. Supplementary Movie 1 shows the diatropic ring current of **5′** on the norcorrole plane and the spatial current stream between the two norcorrole planes. Similar features of the current stream were also confirmed in the case of the dimer with linkers **5** (Supplementary Fig. 17c and Supplementary Movie 2). Such complex through-space current channels have not yet been observed in conventional aromatic molecules. The through-space interplanar current stream is also evident from the anisotropy of the induced current density (ACID) plots (Supplementary Fig. 18)[26]. The observed three-dimensional current pattern can be rationalized in terms of strong intramolecular interactions in **5**.

**Interactions between the two norcorrole subunits**. The non-distorted planar structure of the norcorrole subunits in **5** implies the existence of an attractive force between these two antiaromatic systems. To investigate the interaction between the two norcorrole subunits, the ultraviolet (UV)/visible/near infrared (NIR) absorption spectra of **5** and **6** were measured in dichloromethane (Fig. 5a). The absorption spectrum of **6** shows distinct bands in the higher energy region (<600 nm) with a tailing feature in the lower energy region (>600 nm). These spectral features are characteristic of antiaromatic porphyrinoids and their optically forbidden transition from the highest occupied molecular orbital (HOMO) to the lowest unoccupied molecular orbital (LUMO)[27]. The absorption spectrum of **5** is markedly different from that of **6**, showing a distinct band at 700–1100 nm. Moreover, compared to the intense band of **6** at 545 nm, the absorption bands of **5** at 436 and 608 nm, with significantly reduced extinction coefficients, reflect a strong interaction between the two stacked norcorroles in **5**[28]. To clarify their electronic structure, the magnetic circular dichroism (MCD) spectra of **5** and **6** were recorded (Supplementary Fig. 6). The MCD spectrum of **6** shows no distinct peak in the lowest-energy band due to its intrashell nature[29]. In contrast, **5** exhibits a distinct MCD signal at ~700 nm. These results indicate that the electronic structure of **5** is completely different from that of common antiaromatic compounds such as **6**.

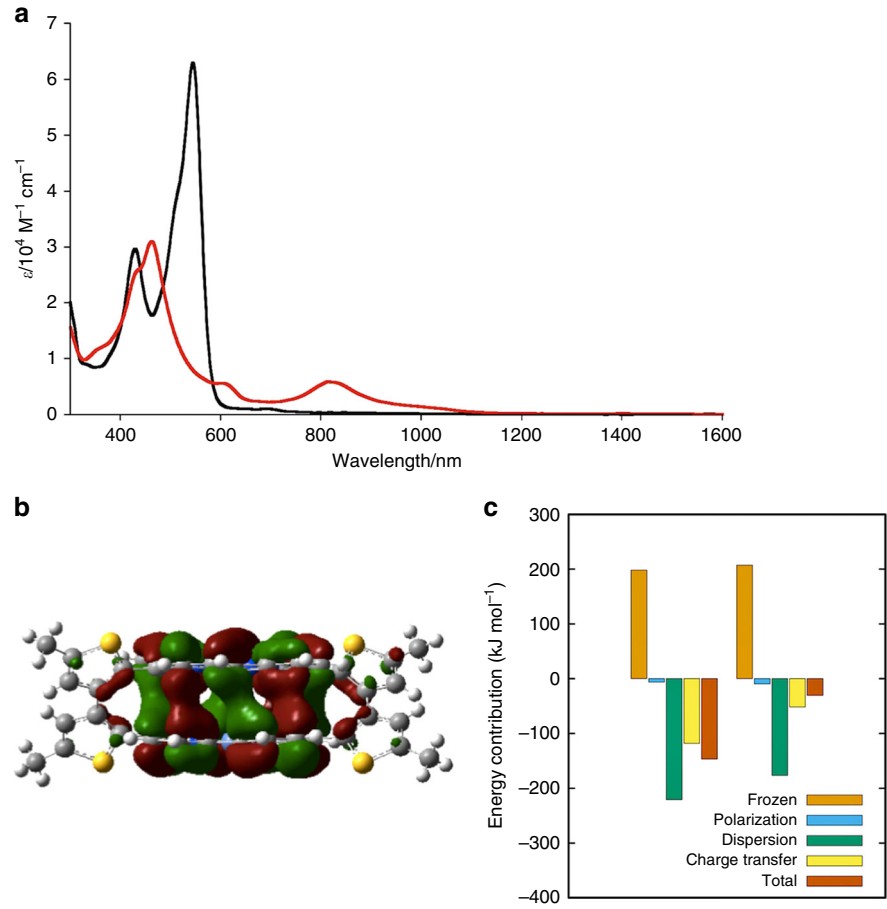

**Fig. 5** Interactions between the two antiaromatic norcorrole subunits. **a** UV/Vis/NIR absorption spectra of **6** (black) and **5** (red) in dichloromethane. **b** The HOMO−1 of **5** calculated at the CAM-B3LYP/6-31G(d) level of theory. **c** Decomposition of the total intermolecular interaction energy in kJ mol⁻¹ of **5′** for the crystal-structure geometry (left) and the crystal-structure geometry without the Ni atoms (right)

The electronic structure of **5** was further investigated by theoretical calculations. Molecular orbitals and their energy levels were estimated from DFT calculations at the CAM-B3LYP/6−31G (d) level of theory. Figure 5b and Supplementary Fig. 11 show the HOMO−1, HOMO, LUMO, and LUMO+1 of **5**. All these molecular orbitals are delocalized over both norcorrole systems. Time-dependent (TD)-DFT calculations revealed that the NIR absorption band of **5** at ~815 nm originates from electronic transitions from quasi-degenerate HOMOs to quasi-degenerate LUMOs (Supplementary Fig. 12 and Supplementary Fig. 13 as well as Supplementary Table 3). This electronic structure stands in sharp contrast to that of conventional antiaromatic compounds such as **6**, which exhibit non-degenerate HOMOs and LUMOs that lead to optically forbidden transitions in the NIR region[30,31]. The configuration interactions between the quasi-degenerate four frontier orbitals in **5** represent a diagnostic feature for typical aromatic porphyrins[27]. The DFT calculations also provided insight into the nature of the Ni–Ni interaction (Supplementary Fig. 11c). The interaction between two Ni $dz^2$ orbitals affords a bonding σ (HOMO−12) and an antibonding σ* (HOMO−4) orbital; however, both orbitals are occupied, thus minimizing the bonding character, which is reflected in the small Wiberg bond index between the two nickel atoms (0.087)[32].

To examine the origin of the attractive interactions between the two norcorrole units, an energy decomposition analysis (EDA)[33] was conducted for dimer **5**, using the simplified model **5′** without the bithiophene linkers (Fig. 5c). An EDA allows decomposing the intermolecular interaction energy between two structures into physically meaningful contributions. Although repulsion exists between the π-electrons, the dimer is stabilized significantly by dispersion interactions. In addition, the stacking of two norcorrole rings is stabilized by charge transfer across the dimer. This charge-transfer stabilization is similar in magnitude to the total interaction energy, illustrating its crucial role in holding the dimer together at such short distance. To investigate the role of Ni in these interactions, we conducted an EDA of a similarly stacked dimer consisting of two metal-free norcorroles. In this case, the charge-transfer contribution was reduced substantially (>50%), suggesting that the Ni atoms play an important role in stabilizing the cyclophane by enhancing the charge-transfer interactions.

Two-photon absorption (TPA) processes, a third-order nonlinear optical phenomenon, are closely associated with π-electron delocalization[34]. We thus measured the TPA cross-section values to investigate the interactions between the stacked norcorroles. Notably, the TPA cross-section value of AN-cyclophane **5** (1590 GM at 1700 nm; Supplementary Fig. 7b) is more than five times that of **6** (270 GM at 1700 nm; Supplementary Fig. 7a). Together with the delocalized molecular orbitals, the higher TPA cross-section value of **5** corroborates the effective delocalization of π-electrons between the two norcorrole subunits. Moreover, considering that an aromatic character leading to effective π-conjugation for energetic stabilization typically enhances the TPA processes[35], this dramatic increase in TPA values supports the notion that the spatial electron delocalization in the AN-cyclophane reflects the aromatic nature of **5**.

For a more comprehensive understanding of the antiaromatic stacking interactions, we also measured the femtosecond transient absorption (fs-TA) spectra of **5** and **6**. While the TA spectra of **5** and **6** show that their excited-state dynamics are dominated by Ni-induced relaxation dynamics (Supplementary Figs. 8 and 9)[36,37], the contrasting TA spectral features of **5** and **6** nicely illustrate the interactions between the norcorrole moieties in AN-cyclophane **5**. The TA spectrum of **6**, shows sharp and derivative-like TA spectral features at 520–650 nm. These spectral features were attributed to relaxation dynamics through (d–d) states of Ni, which has already been observed in other porphyrinoid nickel complexes. On the other hand, the TA spectrum of **5** shows broad excited-state absorption (ESA) bands at 530–770 nm. These broad ESA bands indicate an increased density of electronic states arising from Ni–Ni interactions in **5**, which is consistent with the charge-transfer interactions revealed by the EDA analysis.

## Discussion

AN-cyclophane **5** was synthesized through twofold intramolecular and intermolecular reductive homo-coupling reactions of dipyrrin Ni(II) complex **3**. AN-cyclophane **5** exhibits a structure that is characterized by two closely stacked norcorrole macrocycles (distance: 3.09 Å). Moreover, **5** exhibits a characteristic absorption band in the NIR region, which originates from the transition between quasi-degenerate frontier orbitals and is attributed to mutual interactions between the two macrocyclic subunits. Such orbital interactions also lead to attractive forces between the two antiaromatic systems. An NMR analysis in combination with theoretical calculations suggested that **5** exhibits substantial three-dimensional aromaticity that arises from molecular-orbital interactions between the two π-conjugated systems. Furthermore, we have demonstrated that **5** displays a unique three-dimensional magnetically induced current stream resulting from the aforementioned three-dimensional electron delocalization. The spatial aromaticity presented herein significantly expands the concept of π-conjugated molecules and design guidelines for advanced functional materials.

## Methods

**Synthesis of compounds**. *Norcorrole cyclophane* **5**: To a mixture of **3** (51.8 mg, 51.3 μmol), Ni(cod)$_2$ (69.3 mg, 125 μmol), and 2,2′-bipyridyl (40.2 mg, 125 μmol), dry THF (20 mL) was added. The solution was stirred at 90 °C for 12 h. The mixture was through alumina pad, and evaporated under reduced pressure to solid residue. Purification by silica-gel column chromatography with CHCl$_3$/hexane and recrystallization from CH$_2$Cl$_2$/MeOH afforded **5** in 7% (1.99 mg, 1.87 μmol) as a black solid and a small amount of **6** as a byproduct as a brown solid.

## Data availability

Crystallographic data (CIF files) for **5** and **6** have been deposited with the Cambridge Crystallographic Data Centre as supplementary publications. CCDC 1921391 (**5**) and CCDC 1921390 (**6**) contain the supplementary crystallographic. These data can be obtained free of charge from the Cambridge Crystallographic Data Centre via www.ccdc. cam.ac.uk/data_request/cif. All other data supporting the findings of this study are available within the article and its Supplementary Information.

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

## Acknowledgements

This work was supported by JSPS KAKENHI grants JP26102003, JP15K21721, and JP17H01190 (to H.S.) as well as JP17J09817 (to R.N.). This work was also supported by JSPS A3 Foresight Program. H.S. gratefully acknowledges support from the Murata Science Foundation. T.K. is a Cottrell Scholar of the Research Corporation for Science Advancement and acknowledges support from NSF award CHE-1664674. The research at Yonsei University was supported by the Samsung Science and Technology Foundation under project no. SSTF-BA1402-10. H.F. thanks the Norwegian Research Council through the CoE Hylleraas Centre for Quantum Molecular Sciences (Grant nos. 262695 and 231571/F20) for support. This work has received support from the Norwegian Supercomputing Program (NOTUR) through a computer time grant (Grant no. NN4654K) and the supercomputing resources of the Korea Institute of Science and Technology Information.

## Author contributions

H.S. and D.K. designed and conducted the project and prepared the manuscript. R.N. carried out the synthesis and characterization. I.H. carried out the X-ray diffraction analysis. J.K. and J.O. performed the photophysical measurements and NICS calculations. Y.W. and S.S. measured the MCD spectra. A.L. and T.K. performed the EDA analysis. H.F. carried out the current visualization.

## Additional information

**Competing interests:** The authors declare no competing interests.

