## [Peer Review File · Nature Communications]

Reviewers' comments:

Reviewer #1 (Remarks to the Author):

This work deals with an experimental, in combination with computational, study on the three-dimensional (3D) aromaticity of a cyclophane system with two antiaromatic porphyrin moieties. The topic is of current interest in the literature. Both experimental and computational methodologies are reliable and thus the results should be trustworthy. My only yet major concern is its novelty. The idea of 3D aromaticity is not new (see References 3-6 in the citation of the manuscript). More importantly, 3D aromaticity in antiaromatic cyclophane has been reported in the literature by the very same authors (Ref. 21 of the manuscript). I understand that this work presented a newly synthesized system, which is able to demonstrate even more unambiguously about the existence and validity of 3D aromaticity through the stacking of antiaromatic rings. I also understand that, comparing to Ref. 21, a lot more insights from the computational perspective have been obtained (even though not completely new, I'd say). Nevertheless, all these efforts and new results do not eliminate my severe concern that this work lacks the adequate novelty from the conceptual viewpoint to warrant its consideration for publication on this Journal. In my personal opinion, this work is more suitable to be published in a more specified journal.

Reviewer #2 (Remarks to the Author):

3D aromaticity was predicted theoretically some time ago as a new form of aromatic stabilization as a result of through-space interaction between two antiaromatic networks but has been validated only poorly. The main cause is apparently synthetic difficulty to create such molecules. Prof. Shinokubo challenged this problem and reported an interesting paper on 3D aromaticity (ref. 21) by taking advantage of a stable Ni(II) norcorrole that was developed by his group. Now his group reported a more convincing paper by synthesizing a face-to-face cyclophane type Ni(II) norcorrole dimer. This dimer has been thoroughly studied from the viewpoint of 3D aromaticity and all the experimental data (X-ray structure, ¹H NMR spectrum, NICS calculation, GIMIC calculation, ACID calculation, absorption spectrum, two-photon absorption cross section) indicate the 3D aromaticity of this dimer. As such, this paper is groundbreaking, indicating that 3D aromaticity is really realizable by proper molecular design. This paper is recommended for the publication in Nature Communication as it stands.

Reviewer #3 (Remarks to the Author):

The paper reports results of a second in a row attempt of the research group led by Prof. Shinokubo of a synthesis of three-dimensional aromatic system based on the antiaromatic norcorrole moiety.[1] The present approach involves synthesis of bis(meso,meso-bis(dithiophene))-bridged bis(norcorrolatonickel(II)) where the distance between the macrocyclic rings and their orientations is well-defined by virtue of a two-point bridging and by rigidity of the bridges. According to several experimental results and theoretical calculations, the major goal of the attempt has been reached. The indication of aromaticity of the dimer includes appropriate spectral changes observed in the proton NMR, corroborate by DFT calculations of aromaticity indices (NICS), alteration of the optical properties from typical of antiaromatic macrocycles to those characteristic for aromatic systems, and calculations of diatropic current distribution. Thus, arguments for the aromaticity of the dimeric system **5** are quite convincing. In my opinion that finding, apart from the synthetic success, is the most important achievement of this contribution because here, for the first time, aromaticity of the 3D system consisting of interacting antiaromatic components has been established unequivocally. The paper is clearly written, the results are presented sufficiently detailed and the results are very important, in particularly considering unsuccessful efforts quoted in the paper that have been undertaken previously by this and other groups, following theoretical anticipations of the antiaromatic-to-aromatic alteration. Therefore, in my opinion the data and their interpretation contained in the manuscript and in supporting information, merit publication in Nature Communications, although some point listed below may be addressed or considered by the authors as beneficial for the quality and significance of the paper.

1. The yield of the synthesis of **5** and the total reported outcome of the macrocyclic products are low. What is the fate of the rest of the bis(dibromo-dipyrrin)nickel(II) complex? A word of comment placed in the synthetic part of the Supporting Information section in this regard will be useful.
2. Aromaticity should be reflected also by structural features, in particular by similarity of the bond lengths within the conjugated system. Conversely, it is well-established that for the monomeric antiaromatic norcorroles the bond lengths alternate considerably. There is, however, almost no structural details given in the paper for **5** and no comparison of these between **5** and **6**. I understand that there may be not enough space for such a discussion within the Communication limit, but it can be included in the Supporting Information section. Alternatively, the aromatization may be well indicated by the HOMA [2] index calculated for both **5** and **6** on the basis of either X-ray structures or DFT-optimized models, at least in the planar parts of the dimer.
3. The authors termed the dimeric system as "antiaromatic cyclophane" which is not entirely coherent with the commonly (such as in IUPAC Gold Book) used term "cyclophane" denoting a system consisting of aromatic rings and aliphatic bridges. None of these component is present in **5**. But putting aside the nomenclatural controversies, the character of meso-bridges may be of importance for interpretation of the magnetic properties of this compound. In fact, the π -electronic systems of thiophene rings may be included in the delocalization path involving parts of norcorrole rings with several 24π -electron resonance forms. The possibility of other aromaticity sources than interaction of the antiaromatic rings in **5** may be considered or excluded for this non

planar and skewed system by analysis of its topology.[3] The authors used only one of the possible conformers (chiral, point symmetry D_2) for the calculations of aromaticity indices and, as I understand, the geometries of possible conformers (including D_2 and C_{2v}) were not optimized and the energies of D_2 and C_{2v} geometries were not compared. There is also no indication that solution and solid state most stable structures are of the same twist. It is pity that for GIMICS and EDA calculations a simplified model without bithiophene linkers was used. I partially understand the need for "saving computational costs" but not quite understand the "consistency reasons". These calculations cannot be consistent with the experimental results as the bridges may contribute in both diatropic current distribution and in interactions between the norcorrole rings. In fact, the 3D distribution of NICS (Figure S14) indicates contribution of the bis(thiophene) linkers to the aromatic system.

4. I miss electrochemical data for 5 and 6. The fascinating redox properties of norcorrole nickel(II) complexes (narrow electrochemical HOMO-LUMO gap, high reduction and low oxidation potentials, strong substituent dependence of the potentials) are emblematic for the antiaromatic systems. It is extremely interesting how these properties are changed in the 3D-aromatic system composed of such macrocycles.

5. Minor points:

In several places of the text, the compound identification number **5** should be boldfaced.

p. 4 line 2 under the Figure 3: **3c** instead of **3b** twice?

In ref. 19 the author list should appear as follows: X. Li, Y. Meng, P. Yi, M. Stępień, P. J. Chmielewski

Reference List

1. R. Nozawa, H. Tanaka, W. Cha, Y. Hong, I. Hisaki, S. Shimizu, J.-Y. Shin, T. Kowalczyk, S. Irle, D. Kim, and H. Shinokubo, *Nat. Commun.*, **2016**, *7*:13620, doi: 10.1038/ncomms13620.
2. T. M. Krygowski and M. K. Cyrański, *Chem.Rev.*, **2001**, *101*, 1385.
3. M. Stępień, N. Sprutta, and L. Latos-Grażyński, *Angew.Chem.Int.Ed.*, **2011**, *50*, 4288.

Comments form Reviewer #1

This work deals with an experimental, in combination with computational, study on the three-dimensional (3D) aromaticity of a cyclophane system with two antiaromatic porphyrin moieties. The topic is of current interest in the literature. Both experimental and computational methodologies are reliable and thus the results should be trustworthy. My only yet major concern is its novelty. The idea of 3D aromaticity is not new (see References 3-6 in the citation of the manuscript). More importantly, 3D aromaticity in antiaromatic cyclophane has been reported in the literature by the very same authors (Ref. 21 of the manuscript). I understand that this work presented a newly synthesized system, which is able to demonstrate even more unambiguously about the existence and validity of 3D aromaticity through the stacking of antiaromatic rings. I also understand that, comparing to Ref. 21, a lot more insights from the computational perspective have been obtained (even though not completely new, I'd say). Nevertheless, all these efforts and new results do not eliminate my severe concern that this work lacks the adequate novelty from the conceptual viewpoint to warrant its consideration for publication on this Journal. In my personal opinion, this work is more suitable to be published in a more specified journal.

Our response

First of all, we express our sincere thanks to this reviewer for his/her extensive reading. We appreciate this critical comment form Reviewer 1. The reviewer 1 is certainly correct. The idea of the 3D aromaticity has been theoretically proposed ten years ago. We have also provided experimental evidences for the emergence of the 3D aromaticity in a stacked antiaromatic porphyrin (Ref. 21). However, the stacked antiaromatic porphyrin in the previous report was not a cyclophane. In the present manuscript, we disclose the first synthesis of an antiaromatic cyclophane which exhibits the distinct 3D aromaticity. As we mentioned in the introductory section, antiaromatic cyclophanes have never been prepared to date in spite of many synthetic challenges. Furthermore, the 3D aromaticity in our previous report was not strong enough and the molecule eventually showed antiaromaticity. In sharp contrast, we now demonstrate the distinct 3D aromaticity of the present antiaromatic cyclophane in this paper.

Comments form Reviewer #2

3D aromaticity was predicted theoretically some time ago as a new form of aromatic stabilization as a result of through-space interaction between two antiaromatic networks but has been validated only poorly. The main cause is apparently synthetic difficulty to create such molecules. Prof. Shinokubo challenged this problem and reported an interesting paper on 3D aromaticity (ref. 21) by taking advantage of a stable Ni(II) norcorrole that was developed by his group. Now his group reported a more

convincing paper by synthesizing a face-to-face cyclophane type Ni(II) norcorrole dimer. This dimer has been thoroughly studied from the viewpoint of 3D aromaticity and all the experimental data (X-ray structure, ¹H NMR spectrum, NICS calculation, GIMIC calculation, ACID calculation, absorption spectrum, two-photon absorption cross section) indicate the 3D aromaticity of this dimer. As such, this paper is groundbreaking, indicating that 3D aromaticity is really realizable by proper molecular design. This paper is recommended for the publication in Nature Communication as it stands.

Our response

We express our sincere thanks to this reviewer for his/her extensive reading and we are so delighted to see these supportive comments.

Comments form Reviewer #3

The paper reports results of a second in a row attempt of the research group led by Prof. Shinokubo of a synthesis of three-dimensional aromatic system based on the antiaromatic norcorrole moiety. The present approach involves synthesis of bis(meso,meso-bis(dithiophene))-bridged bis(norcorrolatonickel(II)) where the distance between the macrocyclic rings and their orientations is well-defined by virtue of a two-point bridging and by rigidity of the bridges. According to several experimental results and theoretical calculations, the major goal of the attempt has been reached. The indication of aromaticity of the dimer includes appropriate spectral changes observed in the proton NMR, corroborate by DFT calculations of aromaticity indices (NICS), alteration of the optical properties from typical of antiaromatic macrocycles to those characteristics for aromatic systems, and calculations of diatropic current distribution. Thus, arguments for the aromaticity of the dimeric system **5** are quite convincing. In my opinion that finding, apart from the synthetic success, is the most important achievement of this contribution because here, for the first time, aromaticity of the 3D system consisting of interacting antiaromatic components has been established unequivocally. The paper is clearly written, the results are presented sufficiently detailed and the results are very important, in particularly considering unsuccessful efforts quoted in the paper that have been undertaken previously by this and other groups, following theoretical anticipations of the antiaromatic-to-aromatic alteration. Therefore, in my opinion the data and their interpretation contained in the manuscript and in supporting information, merit publication in Nature Communications, although some point listed below may be addressed or considered by the authors as beneficial for the quality and significance of the paper.

Our response

We express our sincere thanks to this reviewer for his/her extensive reading and we are so delighted at these supportive comments.

1. The yield of the synthesis of **5** and the total reported outcome of the macrocyclic products are low. What is the fate of the rest of the bis(dibromo-dipyrrin)nickel(II) complex? A word of comment placed in the synthetic part of the Supporting Information section in this regard will be useful.

Our response

In fact, a substantial amount of materials was lost during the silica-gel column purification. We detected a trimeric product by the mass spectrometry analysis but we could not characterize it. We consider that oligomeric products were formed in the reaction but such oligomers were not eluted out from silica-gel. We also found that di(4-bromo-5-methyl-2-thienyl)norcorrole **4** was not stable and rapidly decomposed under ambient conditions. The decomposition products also remained in the silica-gel column. We have included a comment on this matter in the experimental section.

2. Aromaticity should be reflected also by structural features, in particular by similarity of the bond lengths within the conjugated system. Conversely, it is well-established that for the monomeric antiaromatic norcorroles the bond lengths alternate considerably. There is, however, almost no structural details given in the paper for **5** and no comparison of these between **5** and **6**. I understand that there may be not enough space for such a discussion within the Communication limit, but it can be included in the Supporting Information section. Alternatively, the aromatization may be well indicated by the HOMA index calculated for both **5** and **6** on the basis of either X-ray structures or DFT-optimized models, at least in the planar parts of the dimer.

Our response

We thank Reviewer #3 for this useful comment. According to his/her suggestion, we evaluated the HOMA values of **5** and **6** on the basis of their X-ray structures. The HOMA values of two norcorrole ring in **5** (0.57 and 0.54) are higher than that in **6** (0.38). This result support the more effective electron delocalization in the cyclophane **5** than the monomer **6**. We have included the discussion of the bond length alternation based on the HOMA values as follows.

Interestingly, the bond length alternation (BLA) in cyclophane **5** is significantly smaller than that in monomer **6**, indicating the more effective delocalization π -electrons in **5**. The BLA can be evaluated

using the harmonic oscillator model of aromaticity (HOMA) values. While a HOMA value of **6** is 0.38, those of two norcorrole rings in **5** are 0.57 and 0.54.

3. The authors termed the dimeric system as “antiaromatic cyclophane“ which is not entirely coherent with the commonly (such as in IUPAC Gold Book) used term “cyclophane“ denoting a system consisting of aromatic rings and aliphatic bridges. None of these components is present in **5**.

Our response

We appreciate the comment from this reviewer. It is true that the term “cyclophane” denotes a macrocycle containing aromatic rings. However, we cannot find a better term to describe our molecule, which is a macrocycle containing antiaromatic rings. In addition, the term “cyclophane” has been also used to describe cyclic molecules with antiaromatic subunits in several previous researches (Ref. 3, 4, 8, 9, 11, 12).

But putting aside the nomenclatural controversies, the character of meso-bridges may be of importance for interpretation of the magnetic properties of this compound. In fact, the π -electronic systems of thiophene rings may be included in the delocalization path involving parts of norcorrole rings with several 24π -electron resonance forms. The possibility of other aromaticity sources than interaction of the antiaromatic rings in **5** may be considered or excluded for this nonplanar and skewed system by analysis of its topology.

Our response

This is a truly interesting idea that we may have the Möbius aromaticity in this system. We have not even noticed this possibility. We really appreciate this suggestion by this reviewer. Then we have re-examined the results of NICS, GIMIC, and ACID calculations. The GIMIC analysis on the real system with linkers clearly indicated the main stream of the ring current does not go through the linkers (Movie S2). In addition, NICS plots for the model dimer without the linker and the real molecule show no big difference, indicating that the linkers are not essential for the generation of the aromaticity in the dimer **5**.

The authors used only one of the possible conformers (chiral, point symmetry D_2) for the calculations of aromaticity indices and, as I understand, the geometries of possible conformers (including D_2 and C_{2v}) were not optimized and the energies of D_2 and C_{2v} geometries were not compared. There is also no indication that solution and solid state most stable structures are of the same twist.

Our response

According to the reviewer's suggestion, we have conducted the DFT optimization of other conformers. We found two other conformers than the D_2 structure. However, these conformers are energetically unfavorable than the D_2 conformer by 14.1 and 18.3 kcal mol⁻¹ and they are not likely present in solution. We have included the discussion on the possible conformations of the dimer **5** as follows.

We also investigated the conformations of **5** using density functional theory (DFT) calculations. While **5** in the solid state adopted D_2 conformation, the relative energies of other possible C_2 conformers were much higher.

It is pity that for GIMIC and EDA calculations a simplified model without bithiophene linkers was used. I partially understand the need for "saving computational costs" but not quite understand the "consistency reasons". These calculations cannot be consistent with the experimental results as the bridges may contribute in both diatropic current distribution and in interactions between the norcorrole rings. In fact, the 3D distribution of NICS (Figure S14) indicates contribution of the bis(thiophene) linkers to the aromatic system.

Our response

The EDA calculation can only evaluate the interactions between two independent systems. That is the reason why we had to use the model system without bithiophene linkers. Then we simply used the same model molecule for the GIMIC analysis. The "consistency reasons" is not scientifically important for the GIMIC analysis. Consequently, we have removed the "consistency reasons" in the revised manuscript. Furthermore, we have conducted again the GIMIC analysis for the real system with linkers. The results were in good agreement with the case of the model system. We have included the results in Supplementary Information.

To examine the role of the bithiophene linkers in **5** to generate the 3D aromaticity, we compared the NICS values of the real molecule **5** and the model system **5'** without the linkers. NICS calculation of **5'** exhibits negative value which stands for the diatropic ring current without the bithiophene linkers. The two-dimensional NICS scan of **5'** also demonstrates the same central negative region as in **5**, indicating a negligible contribution of the bithiophene linkers to induce the 3D aromaticity from antiaromatic units.

Figure 4 in the text

NICS Scan without meso-bridges (5')

4. I miss electrochemical data for **5** and **6**. The fascinating redox properties of norcorrole nickel(II) complexes (narrow electrochemical HOMO-LUMO gap, high reduction and low oxidation potentials, strong substituent dependence of the potentials) are emblematic for the antiaromatic systems. It is extremely interesting how these properties are changed in the 3D-aromatic system composed of such macrocycles.

Our response

We completely agree with this reviewer for the importance of electrochemical data for **5** and **6**. Unfortunately, however, we could not obtain meaningful cyclic voltammogram data for these compounds due to their low solubility even after several trials.

5. Minor points:

In several places of the text, the compound identification number **5** should be boldfaced.

p. 4 line 2 under the Figure 3: 3c instead of 3b twice?

In ref. 19 the author list should appear as follows: X. Li, Y. Meng, P. Yi, M. Stępień, P. J. Chmielewski

Our response

We thank Reviewer #3 for his/her careful checking. These typos and errors have been corrected.

REVIEWERS' COMMENTS:

Reviewer #3 (Remarks to the Author):

All my previous concerns and questions are addressed properly in the new version of the manuscript, supporting information, or rebuttal letter. Therefore I recommend publication of the revised paper in the current form.